# Cloud Screening Method in Complex Background Areas Containing Snow and Ice Based on Landsat 9 Images

**DOI:** 10.3390/ijerph192013267

**Published:** 2022-10-14

**Authors:** Tingting Wu, Qing Liu, Ying Jing

**Affiliations:** 1School of Geomatics, Anhui University of Science and Technology, Huainan 232001, China; 2School of Mechanical and Electrical Engineering, Huainan Normal University, Huainan 232038, China; 3Business School, NingboTech University, Ningbo 315100, China

**Keywords:** cloud screening, complex background areas, snow/ice, mixture tuned matched filtering, improved mixed monolithic sieving model

## Abstract

The first step in the application of Landsat 9 imagery is cloud screening, and the International Satellite Cloud Climatology Project (ISCCP) has made cloud screening an important part of the World Climate Research Program. The accurate identification of clouds in remote sensing satellite images containing snow and ice on the subsurface has been a challenging task in the cloud screening process. It is imperative to fully exploit the characteristic heterogeneous information of the cloud and snow subsurface, to solve the problem of cloud–snow confusion in the snow and ice environment, and to carry out research on cloud screening technology without interference from the snow and ice subsurface. In view of this, this paper will systematically carry out research on cloud screening methods in snow and ice environments. In this paper, we propose the building of a cloud screening algorithm that takes into account the difficulty of the fact that snow and ice on the subsurface can easily interfere with cloud recognition, and the influence of an empirical threshold or statistical threshold that makes its application less effective, and then establish a dynamic threshold cloud screening algorithm that is suitable for snow and ice environments. The research results will provide new ideas and perspectives to solve the problem of surface-type interference that most of the existing cloud screening algorithms contend with.

## 1. Introduction

Research on cloud screening has been ongoing and intensive since the advent of satellite imagery, and for Landsat 9 data, there are few studies of cloud screening in the complex background based on this data. The International Satellite Cloud Climatology Program (ISCCP) has conducted an in-depth study of the average cloud coverage that encompasses a global scale, and its final result demonstrates an average cloud coverage of about 70% [1]. The following figure shows the global cloud coverage and snow coverage for November 2021 and January 2022, provided by NASA, where the white areas corresponding to the value 1.0 represent areas completely covered by clouds or snow, and the areas corresponding to the value 0.0 represent cloud-free clear sky areas or snow-free areas. As shown in Figure 1, the coverage of both clouds and snow/ice is very extensive globally, mainly concentrated in the middle and high latitudes. Cloud screening, as a key technology in the process of applying Landsat 9 imagery, has become a research hotspot in the field of meteorology and remote sensing. According to the different basic theories, cloud and snow separation methods can be divided into four categories: cloud screening methods based on threshold, cloud screening methods based on texture features, cloud screening methods based on pattern recognition, and other methods.

### 1.1. Cloud Screening Methods Based on Threshold

The threshold-based cloud screening method aims to set appropriate thresholds by analyzing the reflectance and brightness temperature values of clouds. Since the threshold method is simple and easy to implement, the current cloud screening method is mainly based on the threshold method, but it depends on the threshold setting and the extraction effect will be disturbed by the subsurface [2,3,4]. The short-wave infrared band is widely used in cloud screening because the reflectivity of snow in this spectral range is usually lower than that of clouds. Some studies have synthetically tested the effect of threshold cloud screening in this band and finally determined whether each image element is a cloud image element one by one [5], and this method is suitable for low- and mid-latitude regions where the subsurface contains vegetation, ocean, and other backgrounds. The thermal infrared band is also commonly used to extract cloud body information, and the temperature characteristics of clouds in the thermal infrared band are counted and analyzed; the visible band and thermal infrared band characteristics are combined, and the corresponding threshold range is set for them to extract cloud body information, and this method has achieved better cloud screening results. However, the use of fixed empirical thresholds or statistical thresholds often leads to the low generalizability of cloud screening algorithms [6]. Zhu et al. [7] developed the Fmask (Function of mask) method for cloud, cloud shadow, and snow detection from Landsat images and found that the average overall accuracy of Fmask was as high as 89.0%. However, Fmask may fail to detect clouds in images showing heterogeneous surface reflectance, because the algorithm uses a scene-based threshold and applies the same threshold to all pixels in the image [8]. Wang et al. [9] proposed a temporal-sequence cloud–snow-distinguishing algorithm based on the high-frequency observation characteristics of the Himawari-8 geostationary meteorological satellite. In the MODIS snow cover products, the cloud pixels with NDSI variance lower than a threshold are identified as cloud-free areas and attributed their raw NDSI value, while the snow pixels with NDSI variance greater than the threshold are marked as clouds.

Thresholding methods are the most widely used because they are simple, efficient, and easy to implement, and they identify cloud information by using the difference between cloud and subsurface features. Although each of these methods has achieved certain results, they can only give a relatively coarse classification result when the image is located in a complex terrain area or the subsurface contains snow and ice [10]. The threshold method helps to reduce misclassification to some extent, but it is difficult to accurately determine the threshold, and so achieving high precision remains challenging [4].

### 1.2. Cloud Screening Methods Based on Texture Features

With the improvement of remote sensing image details, the richer information presented by the features, the clearer textures that can be observed, and the more significant spatial variations in image radiance due to the presence of clouds, the use of spatial texture information for cloud screening has become an effective method [11,12].

Most of the existing texture-based algorithms are applied in the cloud extraction of satellite images without snow or snow-covered satellite images without clouds [13,14]. Compared with clouds, snow has more texture, and the presence of some impurities in the snow, such as rocks and vegetation, can lead to some randomness in the texture of the snow. However, when the snow layer is thicker, the ground surface is completely covered, which leads to less texture. Furthermore, clouds, especially thin clouds, are easily disturbed by the subsurface, which can also cause the texture to change. In addition, using more texture features also leads to higher computational effort and cost. Therefore, the method still has many uncertainties and poor adaptability [10]. The separation of clouds and snow using texture features is still challenging.

### 1.3. Cloud Screening Methods Based on Pattern Recognition

The pattern recognition method can describe and integrate the spectral and spatial information within the object, thus forming an effective cloud screening mechanism. In the process of cloud and snow separation, firstly, suitable features should be selected to describe the characteristics and differences of clouds and subsurface through their features, and then suitable classifiers should be selected according to the classification. For example, Tan et al. [15] used the Probabilistic Latent Semantic Analysis (PLSA) thematic model for cloud screening and designed some simple methods for discriminating cloud-like features such as snow and ice. However, this method is time-consuming in the two iterative optimization algorithms. In addition, cirrus clouds with fine textures were recognized as non-cloud areas due to the fact that cloud regions were designated in this method as having little detail. Yuan et al. [16] proposed an object-oriented and bag-of-words (BOW) model for the remote sensing image cloud extraction method, and then applying the BOW model to form dense local features into a tight feature vector. However, most of the pattern recognition methods have not been comprehensively designed for snow and ice in terms of feature selection and classification strategies [17], and further research is still needed in cloud screening.

### 1.4. Current Status of Research on Cloud Screening Based on Other Methods

Three other types of methods are included in this study of cloud screening in satellite images: (a) The method of using multi-temporal image data. This method compares cloudy images with reference images without clouds in different time phases in the same region to extract cloudy areas, and this method often achieves good detection results. Bian et al. [18] investigated cloud screening methods using multi-temporal images for Environment-1 satellite data, using spectral information to extract the target of interest and combining texture information to optimize the extraction results. Eventually, better cloud screening results were obtained, but such methods require satellites with short revisit cycles. (b) The method of using stereo vision image data. The method of cloud screening using stereo visual image data is based on the knowledge that clouds are usually separated from the ground surface. For optical satellite images with stereo imaging capability, Wu et al. [19] obtained M-DSM through the dense matching of stereo images, compared the generated M-DSM with the reference DEM to obtain the seed points of suspected cloud areas on the image, and then combined it with gray information to obtain the final cloud detection results. (c) The method of using deep learning. Remote sensing images can be combined with deep learning to achieve the technology of image classification and feature extraction, and theoretically, this method can result in better cloud recognition accuracy compared with traditional methods [20]. In 2019, the UNet network was introduced into the cloud screening algorithm, and the semantic segmentation of remote sensing images containing clouds was carried out by the UNet network, and UNet used an encoder–decoder structure to obtain finer cloud masks, which provided a new idea for remote sensing image cloud screening [21]. The SegNet network structure has also been successfully applied in the cloud screening process, which improves the shortcomings of the traditional algorithm sensitive to noise by retaining the index of the encoder downsampling, thus outputting a fine upsampling feature map, which is more effective for thick cloud screening [22]. To further improve the feature learning capability, some scholars have constructed multi-scale feature fusion modules, such as the multi-scale convolution and fusion of depth feature maps [23], or the fusion of feature maps at each scale of the decoder [24]. Xia et al. [25] improved the deep residual network with multidimensional input for the cloud/snow recognition. The accuracy of the classification is higher than the traditional support vector machine, random forest, convolution neural networks, and multi-grained cascaded forest. Guo et al. [26] proposed a cloud detection neural network with an encoder–decoder structure, named CDnetV2. CDnetV2 is characterized by an adaptive feature fusing model (AFFM) and high-level semantic information guidance flows (HSIGFs). CDnetV2 can fully utilize features extracted from encoder layers and yield accurate cloud detection results. Experimental results on the ZY-3 data set demonstrate that the proposed CDnetV2 achieves accurate detection accuracy and outperforms several state-of-the-art methods. Although CDnetV2 achieves satisfactory cloud detection results, there are still some errors in thin cloud and fine particle size cloud masks. Chen et al. [27] presented an automatic cloud detection neural network (ACD net) to screen cloud from high-resolution imagery under cloud–snow coexistence. The ACD net consisted of feature extraction networks and cloud boundary refinement module. The feature extraction networks module was designed to extract the cloud, and the cloud boundary refinement module is used to further improve the accuracy of cloud detection. The results showed that the ACD net can provide a reliable cloud detection result in cloud–snow coexistence scenes. However, the ACD net was conducted with the high-resolution Gaofen-1 image data and cannot be applied to Landsat images. In addition, the ACD net cannot cope well with areas of thin cloud.

In summary, the first two methods described above have more special data requirements, the use of a special combination of image data to express the characteristics of clouds in a single image is difficult to reflect, and can often achieve better cloud and snow separation results, while it is difficult to carry out cloud and snow separation in complex terrain areas, and related research still needs to be further explored. The current application of deep learning in the field of cloud screening has accumulated a certain amount of base-level research, but thin cloud screening, cloud screening in snow and ice environments, recognition accuracy, and other aspects still need further research. The study in this paper proposes a Mixed Tuned Matched Filtering (MTMF) method and an improved mixed monolithic sieving model to separate clouds from snow and ice in complex background areas. The method aims to extend the previously described techniques, thus improving visual comprehension and enabling accurate interpretation. This method has potential advantages over traditional cloud screening methods in extracting anomalous information.

## 2. Materials and Methods

### 2.1. Proposed Method Framework

The flowchart of the proposed algorithm is illustrated in Figure 2. The algorithm consists of two main steps: Mixed Tuned Matched Filtering and Improved Mixed Monolithic Sieving Model. For the MTMF part, Landsat 9 images containing thin clouds, point clouds, and thick clouds in complex backgrounds are firstly selected as the dataset for this study, and the MF end-member spectral information is selected by pixel averaging, so as to enhance the response of cloud body information while suppressing the response of background information. Then, the end-member matching information is digitized, and the digitized results are input into the Improved Mixed Monolithic Sieving Model, and output sub-distribution mean, sub-distribution weights, standard deviation, threshold range, separation degree, overlap rate, and other parameters are determined. This method can be used to solve the problem of snow being misclassified as cloud, as frequently occurs using traditional detection methods. The algorithm is introduced below in detail.

### 2.2. Study Area and Data Source

In this study, the Landsat 9 Operational Land Imager 2 (OLI-2) imagery were acquired from the EarthExplorer website (https://earthexplorer.usgs.gov, accessed on 16 July 2022). We applied the algorithm to Landsat 9 images containing thin clouds, point clouds, and thick clouds with complex backgrounds as the dataset for this study. Due to the short period since Landsat 9 first acquired the images, the data selected for this paper are limited for this particular study. We selected all bands (including visible and near-infrared bands) in the Landsat 9 images to maximize the band properties and avoid the loss of band information. Although cloud and snow cover show similar reflectance in the satellite images, the true reflectance of cloud and snow cover is still different in each band, which contains some important diagnostic absorption and reflection features of cloud and snow cover; therefore, all bands are selected in the proposed method.

Landsat 9 images are multiresolution data, having bands at 15 and 30 m spatial resolution, and we resample all the 15 m bands to 30 m resolution (Table 1). Due to the limited availability of Landsat 9 satellite data, a total of four images containing complex background areas with different forms of clouds and snow were selected for algorithm testing. As shown in Table 2, the four selected images all have representative characteristics. Image A contains a large amount of point-like clouds, which have small morphology and easily produce the phenomenon of missed detection in the process of cloud screening; meanwhile, the background area in image A contains a large amount of ice and snow, which cover a large, flat area, and the ice and snow in these flat areas easily produce high reflectivity. In addition, the point-like clouds also have high reflectivity, which can easily cause confusion between the smaller morphology of point clouds and the ice and snow in the flat areas of the background. Image B contains a large number of thick clouds, these thick clouds show a clumped distribution, and the cloud layer at the edge of the thick clouds is thin, which can easily cause the problem of the wrong extraction of cloud boundaries in the process of cloud extraction, while the background area in image B contains a large amount of thick snow and ice, and the snow and ice cover a large area. The thick snow and ice in this flat area can easily produce a high reflectivity, and in addition, thick clouds also have high reflectance, which can easily lead to confusion between thick clouds with clumpy distribution and thicker snow and ice in the flat background area with similar spectral features. Image C contains a large number of small point clouds and clump clouds, and there are also cloud shadows around these point clouds, which are very similar to the cloud layers [28], which can easily cause the problem of wrong cloud shadow extraction in the process of cloud extraction. At the same time, the cloud distribution in image C is scattered, and the dense distribution and irregular distribution of coefficients coexist, which causes great difficulties for cloud extraction. The background area of image C contains a small area of snow and ice, but the snow and ice cover overlap with the cloud cover on the satellite image, and this phenomenon of overlapping snow and ice and clouds in the one-dimensional space generates a large number of mixed image elements. Therefore, the overlap of the irregularly distributed point and clump clouds in image C and the ice and snow in the background of the satellite image in the same image poses a great problem for cloud monitoring and easily causes confusion between clouds and snow and ice. Image D contains a large number of thin clouds and small punctiform clouds, and the background area overlapping with the thin clouds is snow and ice in the high mountains. When cloud detection is carried out in the complex terrain of the high mountains under the snow and ice environment, there are many types of mixed snow fields, such as patchy snow areas with different light and darkness due to the lighting conditions and solar azimuth, frozen rivers with irregular and high reflectivity on the subsurface, unstable and multiple snow/non-snow cycles in a hydrological year, etc. Therefore, in this paper, four representative images were selected as test data due to the limited available data.

To more intuitively reflect the phenomenon that clouds and snow are easily confused in the complex background area threshold, the traditional K-means unsupervised classification method is selected here for cloud extraction [29], and the extraction effect is shown in Table 2, where the yellow color corresponds to cloud pixels and the black color is the non-cloud area. A large amount of snow and ice are misjudged as clouds, which also proves the fact that cloud extraction is often obstructed by complex background information.

### 2.3. Methods

#### 2.3.1. Mixture Tuned Matched Filtering (MTMF)

Mixed Tuned Matched Filtering (MTMF) is a remote sensing data analysis method developed based on matched filtering. It is a relatively convenient method for detecting and mapping targets using appropriate remote sensing data. MTMF enhances the contrast of the target background by suppressing the background [30]. This method has attracted many researchers to test it for mapping land cover types (including tree species) from remote sensing images, and has rarely been applied to scenes with snow/ice.

The study focuses on extracting end-members and using them to generate MTMF images, which is an advanced matching filtering method focused on reducing spurious anomalies. MTMF is considered to be useful for detecting targets after excluding knowledge of the background material [31]. The method can also detect targets with low spectral contrast between the target and the background as well as within the target. The MTMF of data transformed using MNF facilitates the estimation and reduction of background noise. Mathematically, in a multivariate dataset with m bands and ‘*x*’ samples:(1)$$F_{i}(x),i=(1,2,3,\ldots ,p)$$

Any data set is a signal fraction consisting of some background noise, due to various sensors and atmospheric causes. Then, the total signal is characterized by:(2)F(x)=A(x)+B(x)
where the signal and noise components are *A*(*x*) and *B*(*x*), respectively. The MNF affine transformation, which uses the conventional shift difference method to whiten the noise, can estimate both additive and multiplicative noise.

MTMF is used to evaluate the abundance of end members, and feasibility scores between 0 and 1 are used to tune abundance images by lowering false positives. In general, the approach is woven into the following processing order: (1) pre-processing the data for preparing the dataset for further processing, (2) data dimensionality reduction (MNF transformation), (3) estimating spectral end members, (4) identifying and comparing end members selected through visual inspection with automated identification and a spectral library, and (5) preparing a material map using MTMF approach (Figure 3). The details of the procedure are explained in the sections below.

MTMF produces a set of rule images with an MTMF score for each pixel associated with each end member. A value of 1 indicates a very high degree of matching [32]. Pixels with MTMF scores greater than the background value are thought to have a high probability of being correctly identified [33]. The method identifies pixels where the end member signal differs statistically from the average background pixels [34]. This technique has the advantage of not requiring the identification of all possible end members in a scene, and it may be more advantageous in detecting sensitive small targets in remotely sensed images with complex backgrounds, e.g., images captured after fires which include small litter or ash. Major spectral features, such as exposed soil or burnt black areas after a fire, make it difficult to discriminate small vegetation fractions in an image. The ability to distinguish small patches of remaining vegetation or ash would provide a better indication of point clouds in remotely sensed images with complex backgrounds.

The training areas were chosen where a typical spectral signature of cloud could be clearly identified for each scene, and they were chosen in relatively flat areas to avoid these illumination effects. This provides a rapid means of detecting specific targets based on matches to specific library or image end member spectrums and does not require knowledge of all the end members within an image scene. MTMF values thresholds can be identified from the fraction maps to show areas with relatively good matches to the end member spectra, as MTMF does not require knowledge of all the end members within the scene [32]. Thus, in areas with snow/ice, where identification of all the end members is difficult, MTMF may be a better choice for classification.

The end member spectra used for this classification were generated from an average of 4 pixels in total that surround the single point defining the training area. This produced more representative spectra than spectra based on a single pixel. Furthermore, the averaging process helped to produce spectra that were less noisy. The bright and dark areas on the DEM (slopes facing the sun-azimuth direction at the time of acquisition and the adjacent poorly illuminated back slopes) were avoided when possible. The procedure was implemented in the software MATLAB 2020b to obtain a new image with brighter pixel values to represent the target component. In order to more intuitively show the effect of MTMF on cloud information enhancement and background information suppression, the original panchromatic band gray images are added in the following figure, and it can be seen that the cloud information and complex background information, especially snow and ice information, have similar indistinguishable grey-scale values. The grey-scale comparison images of the original panchromatic band grey-scale images and the MTMF processed images are presented in Figure 4.

In the panchromatic gray-scale image of image A in Figure 4, most of the background area on the upper left is snow and ice in the flat area, while the lower right corner is mostly dotted clouds with a small form; however, it is obvious that the gray-scale values of clouds and snow are similar in this panchromatic gray-scale image, and it is difficult to distinguish between them. After MTMF processing, it can be seen that the higher brightness value in the MTMF gray-scale images is the cloud information, and the brightness value of the cloud information is significantly enhanced compared with the single-band gray-scale value. The high reflectivity of snow and ice in the flat area is obviously suppressed after MTMF processing. The difference between the gray value of cloud body information and background information increases obviously, which reduces the difficulty of cloud extraction. In the panchromatic gray-scale image B, the thick snow and ice in the flat area at the top and the large thick clouds at the bottom have similar gray-scale features. The dark black areas in the MTMF gray-scale images correspond to the background areas, indicating that the high reflectivity of snow and ice in flat areas is suppressed after MTMF processing, and the difference between the MTMF values of clouds and the background information is significantly increased, which reduces the difficulty of cloud extraction in the thicker snow background areas. The difficulty of cloud extraction in the thicker snow-covered background area is reduced. In the panchromatic gray-scale map of image C, the scattered clouds and the snow and ice in the background overlap, and the gray-scale values of both are very similar, so it is difficult to distinguish them by visual recognition or traditional remote sensing methods. After MTMF processing, it can be seen that the cloud information is higher in the MTMF gray-scale images, and compared with the single-band gray-scale image, the cloud information is obviously enhanced, and the snow and ice information is obviously suppressed, so that the cloud and background information can be well distinguished by the eye without the help of numerical analysis. The northern part of image D is a large area of snow, and the snow is covered by a long stretch of high mountains. The slope direction of the snow covering different locations in the high mountains varies. Under the illumination conditions, the snow on the sunny side will become more reflective, and its higher latitude means the brightness will become brighter, while the snow on the shady side will demonstrate the phenomenon of the light being blocked, i.e., the slopes on the sunny side are bright and the slopes on the shady side are dark [35]. As a result, the snow throughout the high mountains will appear as a patchy pattern of one brighter and one darker patch. This substantially increases the difficulty of cloud extraction. In addition, the outline of clouds in this representative area is not obvious enough, and the clouds in the northern area appear to be misty, which again increases the difficulty of cloud extraction. It is obvious from the MTMF processed image that most of the information with high MTMF values in the figure are clouds, especially the thick clouds in the south. The background information of the overall high brightness state is well suppressed, which reduces the difficulty of cloud extraction.

#### 2.3.2. Improved Mixed Monolithic Sieving Model

When cloud screening is performed in complex terrain areas under snow and ice conditions, it is difficult to separate clouds from snow in alpine complex terrain areas due to a combination of factors, such as patchy snow areas with varying light and darkness due to lighting conditions and solar azimuth, frozen rivers with irregular subsurface and high reflectivity, and unstable snow fields with multiple snow/non-snow cycles in a hydrological year. The presence of clouds and multiple features in complex terrain with snow and ice environments can be understood as a multi-modal distribution with a mixture of information, and mathematically, this multi-modal distribution made by mixing multiple types of feature information can be expressed as a mixed overall sieve. Different normal distributions are one of the most important mathematical features in identifying different features on remote sensing images, and they have an important discriminatory significance. In remote sensing information extraction, most of the feature information obeys normal distribution; even if there are several pieces of feature information that approximately obey or do not obey normal distribution, it is only necessary to analyze the statistical characteristics of the information comprehensively and then perform logarithmic or other types of mathematical transformation to transform it into feature information that obeys normal distribution. The information can be transformed into a normal distribution, and we can therefore consider the cloud information as having a normal distribution which can be described by the mixed monolithic sieving model. In general, the mixed monolithic sieving model of the sample is assumed to be:(3)$$\;P\;(x,\lambda )=\overset{n}{\underset{i=1}{\sum }}a_{i}f\left(x,\lambda_{i}\right)$$
where the parameter set $\lambda =\left(n,\lambda_{1},\ldots ,\lambda_{n},\;a_{1},\ldots ,a_{n}\right)$, *n* represents the number of branches, $f\left(x,\lambda_{i}\right)$ denotes the probability density of the *i*th branch, $\lambda_{i}$ is the corresponding parameter, and $a_{i}$ denotes the weight of the *i*th branch with $\sum_{i=1}^{n}a_{i}=1$, $\;a_{i}$ > 0.

The mixed monolithic sieving model has been applied to quantitative studies of sulfur isotope sources, mineral extraction, and geochemistry, with relatively satisfactory results [36]. Studying this model can extract the information of the features of interest more accurately, and in this paper, the separation degree parameter of each sub-distribution is added to this improved mixed monolithic sieving model.

The separation degree can estimate the probability distribution of each target class based on the representative training data of each target class, and calculate the separability degree between two target classes. Under the assumption of a normal probability distribution, the Bayesian distance *L* can be used as a suitable separability measure. *L* was shown to be a measure of the separability of the misclassification probability from the Bayesian decision rule, and the formula for calculating the Bayesian distance is shown below:(4)$$L=\frac{1}{8}{\left(a_{1}-a_{2}\right)}^{2}\frac{2}{\sigma_{1}^{2}+\sigma_{2}^{2}}+\frac{1}{2}ln\left[\frac{\sigma_{1}^{2}+\sigma_{2}^{2}}{2\sigma_{1}\sigma_{2}}\right]$$
where $a_{1}$ and $a_{2}$ represent the means of the two distribution characteristics, respectively, and $\sigma_{1}^{2}$ and $\sigma_{2}^{2}$ represent the variances of the two distribution characteristics, respectively. The first term in the above equation is zero if the means agree, while the second term is zero if the variances of the two characteristic distributions are equal.

In the case of classification environments containing confusing features, the more effective JM distance class can be used to describe the separability, unlike the *L* separability calculation method described above, which uses the JM distance class to describe the separability to obtain a limited dynamic range interval. This allows a better comparison of feature separation results to identify features with separability. JM distance varies on a scale from 0 to 2 to measure the separability of two classes in terms of *L* parameters:(5)$$\mathrm{JM}=2\left(1-e^{-L}\right)$$

If two sub-distribution features are completely separable, they can be represented by JM = 2. In other words, in the case of JM = 2, it means that the two categories can be completely separated on a feature, and no classification error occurs if the feature is used for classification. The smaller the JM value, the worse the separability, the more confusion in the sub-distribution, and the more the classification is error prone. When JM = 0, this indicates that the two categories are almost completely confused on a particular feature.

In this paper, the following improvements are made based on the improved mixed monolithic sieving model:(a)To overcome the situation where the global threshold classification of the overall data causes the internal structure of the data to be obscured, and the local data can be subdivided or subdivided several times according to the parameters of each sub-distribution in the hybrid distribution, to extract the feature information of interest more accurately.(b)Based on the difference between the extracted target area and its background, the reasonable threshold range of the target extracted information can be calculated automatically, so that each pixel in the satellite image is either the area of information to be extracted or the background area, which reduces the subjectivity of the threshold judgment and improves the efficiency and accuracy of the target feature extraction.(c)To increase the standard deviation of each sub-distribution, which reflects the dispersion degree of the sub-distribution data. The larger the standard deviation is, the more unevenly distributed the feature is; the smaller the standard deviation is, the more uniformly distributed the feature is.(d)To increase the separation of each sub-distribution, which can indicate the degree of differentiation between the features to be extracted and the background information. If the separation is large, the features to be extracted can be easily separated from the background information; if the separation is small, the confusion between the two features is large and they cannot be easily distinguished from each other. The improved approach is computationally implemented on the MATLAB platform, and the data used are the values obtained after MTMF.

#### 2.3.3. Accuracy Assessment

In order to evaluate the cloud extraction effect more objectively, user’s accuracy (UA), producer’s accuracy (PA), and overall accuracy (OA) are adopted for quantitative evaluation. The accuracy assessments are as follows (Equations (6)–(8)):(6)$$\mathrm{UA}=\frac{\mathrm{agreement}\;\mathrm{of}\;\mathrm{cloud}}{\mathrm{agreement}\;\mathrm{of}\;\mathrm{cloud}+\mathrm{commission}\;\mathrm{of}\;\mathrm{cloud}}$$
(7)$$\mathrm{PA}=\frac{\mathrm{agreement}\;\mathrm{of}\;\mathrm{cloud}}{\mathrm{agreement}\;\mathrm{of}\;\mathrm{cloud}+\mathrm{omission}\;\mathrm{of}\;\mathrm{cloud}}$$
(8)$$\mathrm{OA}=\frac{\mathrm{agreement}\;\mathrm{between}\;\mathrm{manual}\;\mathrm{mask}\;\mathrm{and}\;\mathrm{algorithm}\;\mathrm{mask}}{\mathrm{total}\;\mathrm{pixels}}$$

## 3. Results and Discussion

The results of the operation on the MTMF values of the study region are as follows: the overall MTMF values can be divided into two normal distributions, and one normal distribution is generally considered to represent a geological phenomenon or a class of features. The cloud information in the gray-scale image after MTMF is enhanced, and other non-cloud information is weakened. Sub-distribution diagrams of the mixed monolithic sieving model are shown in Figure 5.

The improved mixed monolithic sieving model uses the values obtained by MTMF as the basic data, and the model can be calculated in the MATLAB platform to obtain normal distribution diagrams (Figure 5), while the basic parameters in the following tables (Table 3, Table 4, Table 5 and Table 6) can be obtained after calculating each sub-distribution in the MATLAB. To summarize, the gray-scale image obtained from the MTMF can be divided into two categories, i.e., clouds and non-clouds. Lower pixels in the MTMF gray-scale image indicate no target component, which is the corresponding background value, and pixels with higher MTMF values are considered to contain higher target components, i.e., the cloud information we want to extract. The following tables show the specific information of each sub-distribution in the four images in a quantitative way.

Table 3 shows the parameter values of each sub-distribution for the data of image A after processing by the improved mixed monolithic sieving model. Among them, the normal distribution on the left is normal distribution 1, and the normal distribution on the right is normal distribution 2. The mean value of the data within normal distribution 1 is 15,037.78, and the mean value of the data within normal distribution 2 is 28,165.49. The weights of the left and right sub-distributions are 0.75 and 0.25, respectively, indicating that the background information accounts for slightly more weight than the cloud information, which is caused by the fact that most clouds in the image are point-like clouds with small shapes. The standard deviations of normal distribution 1 and normal distribution 2 are 2715.31 and 3492.23, respectively. The standard deviation reflects the dispersion of the data involved in the mixed overall sieving relative to the overall data mean, which means that the dispersion of the data among individuals within the normal distribution 2 group is more discrete than that of normal distribution 1, and the data are not uniformly distributed, which is caused by the disorderly distribution of small point clouds in normal distribution 2. The thresholds of the two subnormal distributions corresponding to Table 3 are from minimum to 22,971.68 and from 22,971.68 to maximum, where the threshold range is automatically obtained by the algorithm. The separation of the two sub-distributions is 1.78 and 0.89, respectively, indicating that the first sub-distribution is easier to separate from the overall data compared to the second one, and the extraction of cloud body information is relatively difficult. The overlap rates of normal distribution 1 and normal distribution 2 are 17.35% and 21.42%, respectively, which also confirms the cloud–snow confusion phenomenon.

The mean value in the table is the mean value of the data within each normal distribution. The mean values of the two sub-distributions are 15,680.72 and 26,966.67, respectively; the weight is a relative concept established based on a certain object, the weight of a certain indicator represents the importance of the indicator relative to the overall evaluation, and the two weight values are 0.69 and 0.31, respectively, so the background information containing snow and ice accounts for a larger proportion and clouds account for a relatively small proportion. The standard deviation is an indicator used to reflect the dispersion of a set of data relative to the mean. Large standard deviation indicates that a large number of data are different from the mean of the overall data; a small standard deviation indicates that only a small number of data are different from the mean of the overall data, and most of the remaining data are close to the mean. The standard deviations of normal distribution 1 and normal distribution 2 are 2803.01 and 7390.65, respectively, which means that the data of individuals in the second normal distribution group are more discrete than those in the first normal distribution group, and the data are not evenly distributed. The improved mixed monolithic sieving model can automatically derive the threshold range to automatically extract the feature body to be extracted, and the thresholds of the two normal distributions corresponding to the following table are: minimum to 20859.38 and 20859.38 to maximum, respectively. The separation is the degree of differentiation between the features to be extracted and the background information, and the separation of the two sub-distributions are 1.02 and 0.72, respectively, indicating that the second sub-distribution is easier to be separated from the overall data compared to the first one. The overlap rate refers to the ratio of the overlapped data in the sub-distribution to the overall data, and the ratio of the overlapped data in the two sub-distributions to the overall data is 61.18% and 30.79%, respectively.

Table 5 represents the parameter values of each sub-distribution of the improved mixed monolithic sieving model for image C. Among them, the mean distributions of the data for the two normal distributions are 22,526.57 and 31,229.21, the weight of normal distribution 1 is 0.55, and the weight of normal distribution 2 is 0.45, indicating that the weight of background information and cloud body information is similar, and the background information is slightly larger than the cloud body information. The standard deviations of normal distribution 1 and normal distribution 2 are 4491.07 and 6844.61, respectively, and the distribution of clouds in satellite images is discrete and irregular. The dispersion of the data involved in the overall sieving of the mixture is higher than the mean value of the overall data, which means that the dispersion of data among individuals in normal distribution 2 is more discrete than that of normal distribution 1, and the distribution of cloud information data is not uniform. This is due to the large differences in the morphological characteristics and spatial distribution of cloud bodies within normal distribution 2. The thresholds of the two sub-normal distributions corresponding to Table 5 are minimum to 26,712.65 and 26,712.65 to maximum, where the threshold range is automatically given by the algorithm. The separation of the two sub-distributions is 0.56 and 0.33, respectively, indicating that it is relatively difficult to perform cloud information extraction in complex background areas. The overlap rates of normal distribution 1 and normal distribution 2 are 77.76% and 51.02%, respectively, and the superposition of clouds and snow in the image adds difficulty to the extraction of clouds in the complex background information.

The mean values of the data for each normal distribution in Table 6 are 8761.75 and 16,423.35, respectively; the weight values of the two normal distributions are 0.36 and 0.64, respectively, because the northern part of image D contains a large number of thin clouds and the southern part has irregularly distributed punctiform clouds; the standard deviations of normal distribution 1 and normal distribution 2 are 9724.56 and 3352.91, respectively. This is due to the fact that the background information contains snow and ice in the high mountains, cloud separation in the complex terrain of the high mountains under snow and ice conditions, patchy snow areas with varying light and darkness due to light conditions and solar azimuth, frozen rivers with irregular and high reflectance on the subsurface, and the unstable mixture of influences within one hydrological year, such as snowfields with multiple snow/non-snow cycles in a hydrological year, that increase the difference between the background information and the average of the overall data. The improved mixed monolithic sieving model can automatically derive the threshold range to automatically extract the features to be extracted, and the thresholds of the two normal distributions corresponding to Table 6 are: minimum to 9724.56 and 9724.56 to maximum, respectively. The separation is the degree of differentiation between the features to be extracted and the background information. The separation of the two sub-distributions is 1.53 and 1.46, respectively, indicating that the first sub-distribution is more easily separated from the overall data compared to the second one. The ratio of the overlapping data of the two sub-distributions to the overall data is 27.43% and 8.66%, respectively, which is beneficial to the subsequent cloud extraction in high mountainous and complex terrain areas.

In order to verify the effectiveness of this method, we compare the proposed approach with the ANN and Fmask approaches, which are two algorithms with good cloud extraction results [7,20]. Figure 6 shows the four regions of interest with their cloud extraction results. The first column in Figure 6 displays the original images, which cover various surface environments, including bright backgrounds, mountains, ice, and snow. The second column of false color composite images helps the reader visually differentiate between snow/ice and clouds, with bands 7, 4, and 2 for Red, Green, and Blue, respectively. Columns 3–5 are the results of cloud extraction using ANN, Fmask, and the proposed method, where the yellow color represents the extracted cloud information and the black color represents the non-cloud subsurface area. We applied ANN and Fmask to ENVI 5.4 software here. The details are shown in the figure below.

By comparison, the ANN and Fmask methods seem to overestimate the number of clouds, as shown in Figure 6, which also verifies the fact that images with snow and ice in the background are easily misclassified as clouds. Through visual recognition, it can be seen that the method proposed in this paper has a better effect on extracting cloud information and effectively reduces the problem of cloud and snow/ice confusion. The quantification results of the four complex background areas containing both clouds and snow are shown in the Table 7. Three different accuracy assessments were used to assess the accuracy of the algorithm results.

The verification samples (reference mask) were obtained by the visual evaluation of the scene in Adobe Photoshop CS6. The results of the proposed method are better than those of the FMask algorithm and ANN method, especially for UA, which is significantly improved compared with the other two methods. This is because the proposed method effectively improves the phenomenon wherein snow and ice are misjudged as clouds, and reduces the occurrence of cloud and snow confusion. The proposed method achieves high precision in cloud detection in areas where it is difficult to extract cloud. The results of cloud screening using the extracted cloud-covered data indicated that PA exceeds 80%, UA exceeds 80%, and OA exceeds 90%. Therefore, this method can solve the difficult problem wherein background ice and snow are misjudged as clouds, and the problem of point clouds with very small morphology being missed, and it also has better results for cloud–snow separation in complex terrain areas.

## 4. Conclusions

The study focused on cloud screening methods in complex background areas containing snow and ice, based on Landsat 9 Images. We compared the ANN and FMask methods for cloud detection to the proposed technique and concluded the following:(1)Since Landsat 9 data are limited, four representative test images are selected in this paper, and the four images contain different forms of clouds and complex background information at the same time. This paper adopts the mixture tuned matched filtering method, and the research results show that the method has an obvious effect on cloud information enhancement in complex background images containing snow and ice, and at the same time, the method can suppress complex background information in a clear manner.(2)This paper improved a mixed monolithic sieving model, which overcomes the issue of the global threshold classification of the overall data causing the internal structure of the data to be obscured; this method can automatically calculate a reasonable threshold range for the information extracted from the target, so that each pixel in the satellite image is the information region or background region to be extracted, which reduces the subjectivity of threshold judgment and improves the efficiency and accuracy of target feature extraction.(3)The proposed method greatly reduces the misclassification of clouds and snow, it is better for the recognition of point-like clouds with small morphology, and in the case that the background area contains snow on high mountains, the method is also effective without interference from patchy areas generated by snow on high mountains.

The proposed method is not limited by indefinable texture characters, complexity, and other factors; thus, it is versatile. However, since the data are limited, cloud screening and accuracy assessment results for larger range images will be explored in future research.

## Figures and Tables

**Figure 1 ijerph-19-13267-f001:**
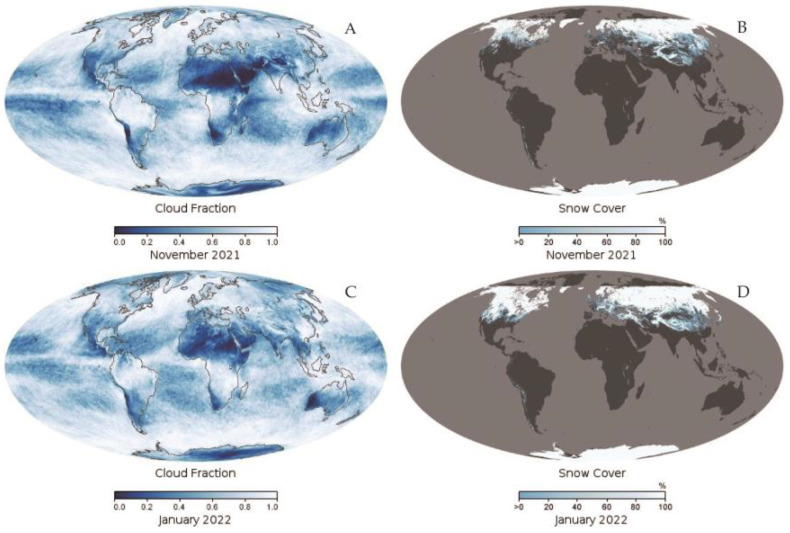
Global cloud cover (**A**,**C**) vs. snow cover (**B**,**D**) in November 2021 and January 2022.

**Figure 2 ijerph-19-13267-f002:**
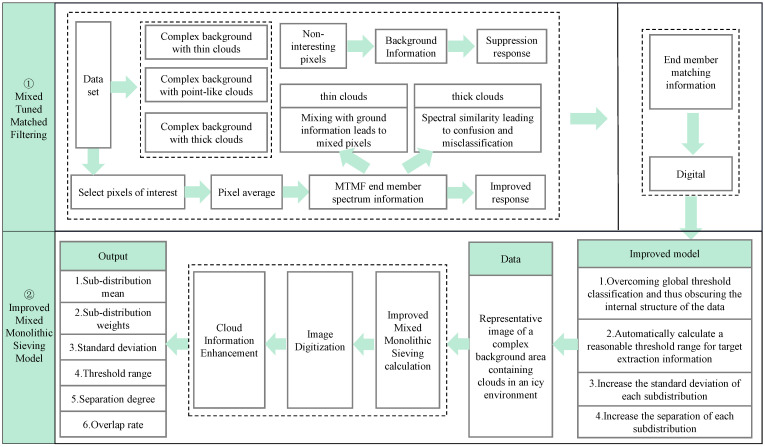
Proposed method framework.

**Figure 3 ijerph-19-13267-f003:**
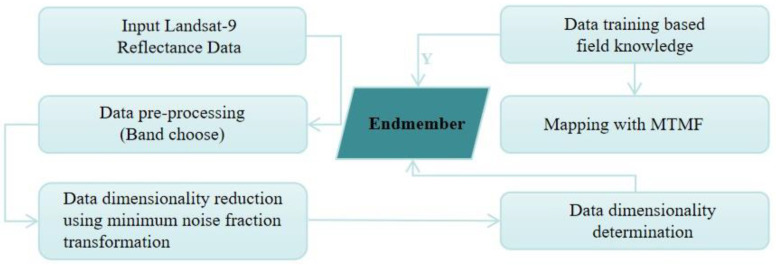
Flow chart of image processing and MTMF-based mapping steps.

**Figure 4 ijerph-19-13267-f004:**
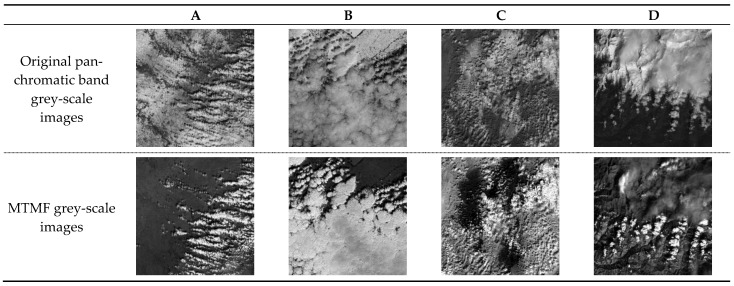
Grey-scale comparison images (**A**–**D**) of the original panchromatic band grey-scale images and the MTMF processed images.

**Figure 5 ijerph-19-13267-f005:**
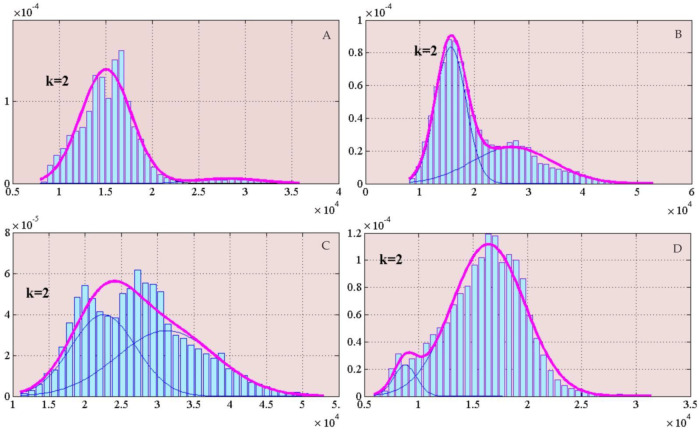
Sub-distribution diagram of the mixed monolithic sieving model, where (**A**–**D**) correspond to the results of the four gray-scale images in Figure 4, respectively.

**Figure 6 ijerph-19-13267-f006:**
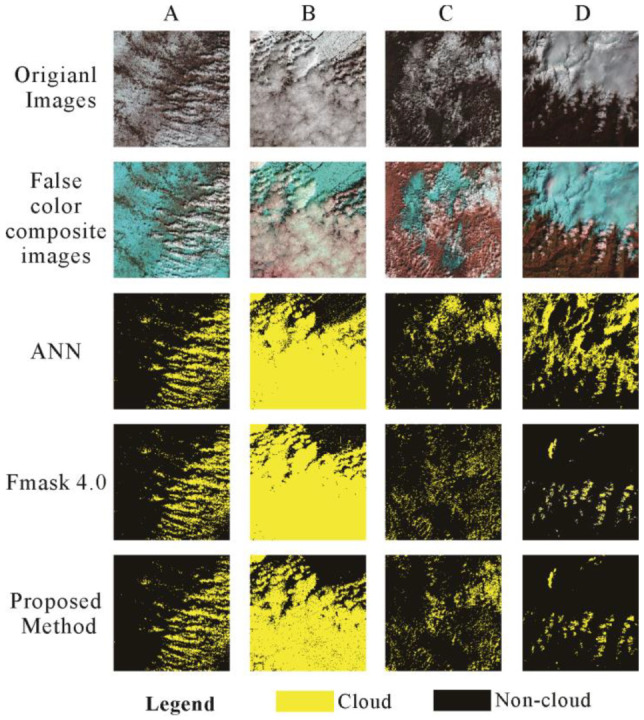
Comparison of the proposed method against ANN and Fmask. (Sub-figures (**A**–**D**) correspond to the four data used in this paper, respectively. Cropping the region of interest, a false color composite is used to help the reader visually differentiate between snow/ice and clouds. Bands 7, 4, and 2 for Red, Green, and Blue, respectively).

**Table 1 ijerph-19-13267-t001:** Details about the OLI-2 spectral bands of Landsat 9.

	Band	Minimum Lower Band Edge (nm)	Maximum Upper Band Edge (nm)	Center Wavelength (nm)	Spatial Resolution (m)
1	Coastal/Aerosol	433	453	443	30
2	Blue	450	515	482	30
3	Green	525	600	562	30
4	Red	630	680	655	30
5	NIR	845	885	865	30
6	SWIR 1	1560	1660	1610	30
7	SWIR 2	2100	2300	2200	30
8	Panchromatic	500	680	590	15
9	Cirrus	1360	1390	1375	30

**Table 2 ijerph-19-13267-t002:** Landsat 9 images with complex background information of the study area (A-D correspond to the four data selected in this paper, respectively. Cropping the region of interest), with the (i) acquired date, (ii) original images, (iii) cloud screening results using the traditional K-means unsupervised classification method.

	A	B	C	D
Acquired Date	11 March 2022	8 March 2022	6 March 2022	4 March 2022
Landsat 9 data (cropping the region of interest)	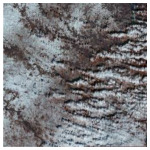	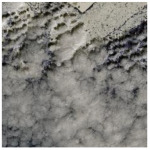	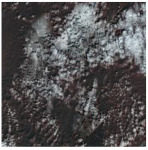	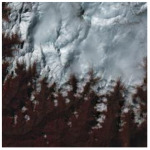
Traditional K-means classification method	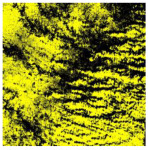	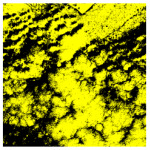	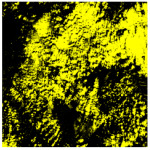	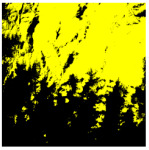

**Table 3 ijerph-19-13267-t003:** Parameter values of each sub-distribution of the improved mixed monolithic sieving model for image A.

	Sub-Distribution 1	Sub-Distribution 2
Sub-distribution mean	15,037.78	28,165.49
Sub-distribution weights	0.75	0.25
Standard deviation	2715.31	3492.23
Threshold range	Minimum~22,971.68	22,971.68~maximum
Separation degree	1.78	0.89
Overlap rate	17.35%	21.42%

**Table 4 ijerph-19-13267-t004:** Parameter values of each sub-distribution of the improved mixed monolithic sieving model for image B.

	Sub-Distribution 1	Sub-Distribution 2
Sub-distribution mean	15,680.72	26,966.67
Sub-distribution weights	0.69	0.31
Standard deviation	2803.01	7390.65
Threshold range	Minimum~20,859.38	20,859.38~maximum
Separation degree	1.02	0.72
Overlap rate	61.18%	30.79%

**Table 5 ijerph-19-13267-t005:** Parameter values of each sub-distribution of the improved mixed monolithic sieving model for image C.

	Sub-Distribution 1	Sub-Distribution 2
Sub-distribution mean	22,526.57	31,229.21
Sub-distribution weights	0.55	0.45
Standard deviation	4491.07	6844.61
Threshold range	Minimum~26,712.65	26,712.65~maximum
Separation degree	0.56	0.33
Overlap rate	77.76%	51.02%

**Table 6 ijerph-19-13267-t006:** Parameter values of each sub-distribution of the improved mixed monolithic sieving model for image D.

	Sub-Distribution 1	Sub-Distribution 2
Sub-distribution mean	8761.75	16,423.35
Sub-distribution weights	0.36	0.64
Standard deviation	4058.53	3352.91
Threshold range	Minimum~9724.56	9724.56~maximum
Separation degree	1.53	1.46
Overlap rate	27.43%	8.66%

**Table 7 ijerph-19-13267-t007:** PA, UA, and OA using the proposed method.

		PA	UA	OA
ANN	A	90.67%	75.84%	94.32%
	B	97.13%	79.94%	78.24%
	C	89.12%	70.14%	94.78%
	D	91.29%	67.38%	94.75%
Fmask	A	89.56%	77.95%	94.69%
	B	97.33%	88.88%	88.14%
	C	82.94%	75.20%	95.25%
	D	87.42%	78.32%	96.35%
Proposed Method	A	88.89%	83.33%	95.71%
	B	96.90%	92.59%	91.33%
	C	88.24%	81.08%	96.54%
	D	81.39%	90.32%	96.99%

## Data Availability

The data that support the findings of this study are openly available in https://earthexplorer.usgs.gov.

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
