# Peer review of "Cloud Screening Method in Complex Background Areas Containing Snow and Ice Based on Landsat 9 Images"

_ijerph, 2022, doi:10.3390/ijerph192013267_

Round 1
Reviewer 1 Report
Dear Author/s
Congratulations on your excellent work. The manuscript 'Cloud Screening Method in Complex Background Areas Containing Snow and Ice Based on Landsat-9 Images' is well written. However, authors should consider some essential comments before publishing the work.
In remote sensing, when an image contains a large part of it under cloud cover, it becomes unusable since the surface below clouds isn't visible and is of no use. However, the efforts to differentiate between could and snow/ice is important for correctly mapping snow/ice. The model used in the present study seems good; however, the authors should first justify the need for such models by answering the following questions
· Are widely used methods already available to differentiate clouds and snow/ice? If yes, what are the accuracy levels of those methods?
· If the SWIR band itself is capable of differentiating clouds and snow, as seen in figure 7 FCC, then why not just use the single SWIR band and use the threshold on it to extract clouds?
· What would be the accuracy difference between this method and the methods used by authors in the present study? If the SWIR band alone can give better results, author/s needs to justify the need for this model in the discussion section.
The overall manuscript explains the study well; however, there are some improvements needed in the manuscript.
· A large part of methodology where results are derived from testing different methods are written should be the part of Results and discussion.
· Results and discussion part are too small; please substantiate with other studies.
· There is no discussion of previous studies or comparison between techniques and limitations of the methods used.
· Map and Figure: Clarity and resolution of the maps are poor. I can't read legends and indexes in figure 1.
· Literature review and use of references are inadequate. Some recently updated references
o need to be added.
· References have to be rechecked as they lack similarity in writing style. In a few references, the page number is written confusingly (Ref 10, 23).
· The authors must check the grammar, consistency and flow of the texts in the manuscript before submitting it to a journal for publication.
Best Wishes!
Reviewer 2 Report
This paper proposes a Mixed Tuned Matched Filtering (MTMF) method and improved mixed monolithic sieving model to separate clouds from snow and ice in complex background areas. This paper explains the proposed method in detai. However, concerning the abstract and introduction, they are poorly designed, and the motivation of this study was not well argued, which made the necessity and innovation of this study weak. There are also some grammatical errors and formatting issues that need to be improved. Some main issues below should be addressed.
1. the author has made too many background descriptions in the abstract, which is not necessary.
2. According to your research, the research progress and difficulties of the threshold method should be emphasized in the introduction. However, a large amount of space is used to introduce the cloud screening method based on texture features and pattern recognition.
3. The proposed method is more inclined to the fusion of multiple existing detection methods and lacks further innovation. In terms of cloud screening (cloud detection) from snow and ice environments, the OA of a new study (DOI: 10.1109/LGRS.2021.3102970) has reached 97%, which seems to have a higher separation effect. And Section 3 lacks the description of the test sample and the comparison experiments. How are the truth values labeled? What is the sample size? To fairly verify the superiority of the proposed method, the authors are suggested to compare the presented method with more competitive baselines.
Reviewer 3 Report
The manuscript has developed a cloud screening algorithm that takes into account the difficulty that snow and ice on the subsurface can easily interfere with cloud recognition, and the accuracy of this method is good with an overall accuracy of 90%. This method could benefit the cloud identification in satellite data, it is meaningful to the relevant researcher. But there still have some problems that need to be revised. Some detailed comments for this paper are as below.
1. Lines 47-49, the sentence needs to be rewritten.
2. Lines 154-160, this sentence is too long, which make it hard to understand, also some grammar issue appear in it. Please check it and rewrite it.
3. Line 393, a more blank character is added between “cloud” and “information”.
4. Please uniform the formula format (size, font, etc.).
5. The structure of this manuscript should be reorganized, and contents related to table 1, table 2, table 3, and table 4 should be added in the result section. Besides, What method was used to obtain the results in table 1, table 2, table 3, and table 4 should be explained more detailedly.
6. Line 478, “The” is miswritten as “Th”.
7. The evaluation method like equation 6, equation 7, and equation 8 should be described in the previous section which mainly describes the methods.
8. The results and discussion section is too short, this section needs to be rewritten.
9. A comparison between the method proposed in this study and other existing methods.
10. The writing of this manuscript needs to be thoroughly checked.
Round 2
Reviewer 1 Report
Dear Author/s
Thanks for incorporating suggestions. I appreciate your efforts in making the manuscripts ready for the scientific community.
Best wishes
Author Response
Thanks so much for whatever has been supported professionally. And we are very appreciated.
Reviewer 2 Report
The revised manuscript has improved significantly. However, before any possible publication, the following comments should get carefully addressed.
1. For your reply to Q2, I think there are still problems with the revision of the introduction, which still contains a lot of discussion that is not very relevant to the content of your study (Sections 1.2 and 1.3). It is recommended that the authors should further reduce the length.
2. For your reply to Q3, the sample I am referring to is not the training sample, but the validation sample used in the comparison experiment (in Table 5).
3. Figure 1, for figures with more than one part, it is recommended to use label “A, B...” or “a, b...” instead of “left, right” for the subfigures. And the meaning of each subfigure should be clearly indicated (e.g. Figure 6).
4. Section 2.2, please add specific information about the bands of Landsat-9 and the time information of the selected images in MATERIALS.
5. There are a lot of problems with the format of formulas and words. For example, in Eq. (1), the variable “i” should be italicized.
6. Section 2.3.3, the evaluation indicators are weakly representative, and in the secondary classification, POD/FAR/CSI, etc. are usually used as the evaluation indicators to verify the accuracy and stability of the algorithm.
Reviewer 3 Report
The manuscript has been improved significantly after a major revision. There still exist some problems in the current version. Some detailed comments for this manuscript are as follows:
1. Figure 2 should be more clear, as its resolution was too low.
2. The brackets in line 354 were in Chinese form, need be modified into English form.
3. The references in line 487, 537, 657, and 933, are in wrong form.
4. The variations in equation 6, 7, and 8 should be represent by letters instead of full name, and the meanings of letters explained after these equations.
